

# Limb-use by foraging marine turtles, an evolutionary perspective

Jessica A. Fujii[1], Don McLeish[2], Andrew J. Brooks[3], John Gaskell[4] and Kyle S. Van Houtan[1,5]

[1] Monterey Bay Aquarium, Monterey, CA, United States of America
[2] Hawaiian Hawksbill Conservation, Lahaina, HI, United States of America
[3] Marine Science Institute, University of California, Santa Barbara, CA, United States of America
[4] Living Reef - Daydream Island, Whitsundays, Queensland, Australia
[5] Nicholas School of the Environment, Duke University, Durham, NC, United States of America

## ABSTRACT

The use of limbs for foraging is documented in both marine and terrestrial tetrapods. These behaviors were once believed to be less likely in marine tetrapods due to the physical constraints of body plans adapted to locomotion in a fluid environment. Despite these obstacles, ten distinct types of limb-use while foraging have been previously reported in nine marine tetrapod families. Here, we expand the types of limb-use documented in marine turtles and put it in context with the diversity of marine tetrapods currently known to use limbs for foraging. Additionally, we suggest that such behaviors could have occurred in ancestral turtles, and thus, possibly extend the evolutionary timeline of limb-use behavior in marine tetrapods back approximately 70 million years. Through direct observation *in situ* and crowd-sourcing, we document the range of behaviors across habitats and prey types, suggesting its widespread occurrence. We argue the presence of these behaviors among marine tetrapods may be limited by limb mobility and evolutionary history, rather than foraging ecology or social learning. These behaviors may also be remnant of ancestral forelimb-use that have been maintained due to a semi-aquatic life history.

# INTRODUCTION

Marine turtles (Chelonioidea Oppel, 1811) and most other marine tetrapods, have evolved body forms that are best suited to move, orient, and minimize drag in a fluid environment rather than using their articulating limbs to directly aid in prey capture or processing (*Fish, 2016*). Due to the limitation of these evolved body plans and the constraints of the aquatic environment, *Taylor (1987)* predicted mouth-based filter, suction, or ram foraging to be the primary foraging mechanisms for all marine tetrapods. Although the evolution of foraging mechanisms generally coincides with associated morphological traits, such as filter feeding and baleen in Mysticete whales (*Deméré et al., 2008*), many species have been observed using innovative strategies counter to what their evolved body plans would predict. Following *Gould & Vrba (1982)* and *Lloyd & Gould (2017)*, these traits could be considered exaptations; "traits that were adapted for one evolutionary function, but were

Corresponding author
Jessica A. Fujii, jfujii@mbayaq.org

later co-opted (but not selected) to serve a different role". Such exaptations can provide insight into an organism's current ecological dynamics (*Gould & Vrba, 1982*) as well as the evolutionary conditions influencing these novel behaviors.

Despite the predictions of *Taylor (1987)*, a number of marine tetrapods have been documented to use their limbs to directly aid in prey capture, manipulation, and processing (*Iwaniuk & Whishaw, 2000*). Rudimentary limb-use for foraging is observed in a range of terrestrial and aquatic taxa, therefore, this behavior could have evolved in ancestral tetrapods and was subsequently developed, maintained, or lost in different lineages over time (*Iwaniuk & Whishaw, 2000*). For those lineages that lost the ability, the use of limbs to aid in foraging may be an exaptation—wherein limbs evolved for locomotion have been co-opted to be used in food handling (*Gould & Vrba, 1982*). These behaviors (hereafter "limb-use") may improve foraging efficiency, expand ecological niches, and perhaps confer greater resiliency in dynamic or altered environments. Limb-use could also have developed as a result of co-evolution or are secondary adaptations (*Lloyd & Gould, 2017*), but as suggested by *Hocking et al. (2017b)*, the evolution of forelimbs dedicated to locomotion may have consequently made said limbs unsuitable for feeding purposes. Thus, it seems unlikely that these behaviors would have been selected for under various evolutionary processes.

Why limb-use develops in some marine tetrapods, but not others, is not well understood. *Hocking et al. (2017b)* showed that less-specialized, semi-aquatic marine mammals (Mustelidae, Odobenidae, Otariidae, Phocidae) might retain the use of forelimbs to manipulate prey, but older taxa and taxa more specialized for the aquatic environment (e.g., cetaceans) might rely solely on suction, filter, or ram foraging. Whether this pattern extends to marine turtles has not been previously explored.

Marine turtles are the oldest extant line of marine tetrapods but some still maintain a semi-aquatic lifestyle for thermoregulation and breeding (*Kelley & Pyenson, 2015*). Like many other marine tetrapods, marine turtles predominantly use suction or bite-and-tear foraging strategies to capture and process food (*Moreno et al., 2016*). To date, however, direct observations of marine turtle foraging mechanisms are somewhat limited, or the methods used (such as critter-cams) limit the ability to assess limb-use due to a narrow field of view (but see: *Schofield et al., 2006*; *Seminoff, Jones & Marshall, 2006*; *Patel et al., 2016*). While knowledge of marine turtle diets has significantly improved in recent years with technological innovations (*Arthur et al., 2007*; *Patel et al., 2016*; *Van Houtan et al., 2016*) without direct observations, many aspects of feeding behavior remain overlooked.

Here, we describe three marine turtle species—green (*Chelonia mydas*), hawksbill (*Eretmochelys imbricata*), and loggerhead (*Caretta caretta*)—using limbs in the wild to aid in foraging in ways which have not been previously assessed in detail. We set these observations in context with other marine tetrapods known to use their flippers, forelimbs, or tails as direct aids in obtaining or processing food and discuss the role of behavioral, morphological and ecological factors that may limit or promote this behavior.

**Table 1** Functional definitions of observed types of limb use by feeding marine tetrapods.

| Behavior | Feeding stage[a] | Definition |
| --- | --- | --- |
| Digging | Capture | Using one or both flippers or paws to remove benthic sediment in order to access benthic food. |
| Striking | Capture | Using one or both flippers, or tail, to forcibly hit prey, usually to stun. |
| Tossing | Capture | Using flipper or tail to project prey into the air, usually used to stun prey. |
| Kerplunking | Capture | Slapping water surface with tail to cause a startle response in prey to aid in capture. |
| Leveraging | Processing | Placing one or both flippers against benthic substrate to create tension while pulling food from substrate with mouth. |
| Swiping | Processing | Moving one flipper against food to create tension while tearing food into smaller pieces with mouth. |
| Holding | Processing | Using both flippers to keep food in place, either by squeezing flippers or gripping with claws while pulling food apart with mouth. |
| Pounding | Processing | Using both flippers or paws to hold food while rapidly hitting against another object. |
| Corralling | Transport | Using one or both flippers to guide loose food in a directed manner toward mouth. |
| Lobtailing | Transport | Slapping water surface with tail during bubble-net feeding to corral prey together. |

**Notes.**

[a] Feeding behaviors fell in one of three categories of feeding stages: capture, processing, and transport based on *Hocking et al. (2017b)*.

## MATERIALS & METHODS

While viewing a fixed-station underwater video from a coral reef in Moorea, French Polynesia we opportunistically observed a hawksbill sea turtle use its limbs while foraging, prompting discussion with experts in the field, and a broader survey for the occurrence and context of this behavior. We documented marine turtle foraging behavior from underwater surveys, web image and video searches (e.g., Google, YouTube, Vimeo, Flickr, Shutterstock), and the published literature.

For this study, limb-use for feeding was defined as the intentional use of flippers, paws, tails, or feet to directly aid in the capture, processing, or transport of the animal's food while in the marine environment. If we did not find limb-use feeding described in the published literature (e.g., searching Google Scholar and Web of Science), we conducted broader internet searches for video and images using the species' common name or group ("green turtle" or "sea turtle") combined with feeding terms (e.g., "feeding", "foraging", "eating"). Once an initial record was found, we conducted more in-depth searches for that species or group to determine the ecological context of the behaviors. Feeding strategies were grouped into broad behavioral categories and feeding stages based on previous study definitions (*Hocking et al., 2017b*), Table 1.

We defined marine tetrapods similar to previous studies (*Kelley & Motani, 2015*; *Kelley & Pyenson, 2015*). We excluded the polar bear (*Ursus maritimus*) as there is significant genetic admixture with a fully terrestrial species (*Miller et al., 2012*), and we excluded

marine snakes as they lack external limbs. We initially included sea birds, yet limited observations to feeding occurring entirely in the marine environment. Foot-paddling, for example, is observed in a number of Laridae gulls, yet it occurs in terrestrial or mudflat habitats (*Tinbergen, 1962*) and so was not included.

Due the difficulty of observing wild foraging behaviors for many marine tetrapods, the absence of documented limb-use while feeding here does not indicate the behavior does not or cannot occur. In light of this, our intent was to be descriptive, not exhaustive, in comparing the occurrence of these behaviors. Due to the relative rarity of this behavior, we grouped marine tetrapods into taxonomic families for comparisons. We broadly compared evolutionary, morphological, ecological, and behavioral factors to qualitatively determine if the presence of limb-use behaviors followed patterns across marine tetrapod families. Evolutionary relationships and divergence times are from Timetree.org (*Hedges, Dudley & Kumar, 2006*).

## RESULTS & DISCUSSION

*Carr (1967)* described hatch-year green turtles using the sharp claw on their foreflippers to swipe and tear food in tanks in captivity. *Davenport & Clough (1985)* similarly observed these behaviors in captive juvenile loggerhead turtles. Both studies suggested these behaviors would be limited to juveniles due to undeveloped, weak jaws. Since then, adult loggerhead turtles have been recorded holding mollusks between their foreflippers (*Houghton, Woolmer & Hays, 2000*), and digging (using forelimbs to remove benthic sediment) has been opportunistically documented in adult green turtles (*Christianen et al., 2014*) and loggerhead turtles (*Limpus, Couper & Read, 2001*; *Preen, 1996*). In our surveys, we found additional forms of limb-use not previously described, and add hawksbill turtles to the species of marine turtles documented to use these feeding strategies.

We found four novel types of limb-use by hawksbill, loggerhead, and green turtles. Hawksbill and green turtles were observed using corralling, leveraging, holding, and swiping movements to capture, process, or transport a variety of sponges, cnidarians, macroalgae, and fishes (Figs. 1A–1D, 1F, Fig. S1, Table 1). We also documented loggerhead sea turtles swiping to process benthic mollusks, which is likely a continuation of the holding behaviors described above (Fig. 1E). Given the apparent rarity of these feeding strategies, it seems unlikely that they are required to consume any of these prey items, but they may aid feeding efficiency and expand foraging or habitat niches.

Limb-use while feeding has been previously reported in eight additional families of marine tetrapods including Balaenopteridae, Delphinidae, Trichechidae, Dugongidae, Mustelidae, Odobenidae, Otariidae, and Phocidae (Fig. 2, Table S1). Within these families, ten types of limb-use for foraging have been observed: digging, striking, tossing, kerplunking, leveraging, swiping, holding, pounding, lobtailling, and corralling (Table 1). Holding and digging were the most common behaviors seen across families (*Kastelein & Mosterd, 1989*; *Bowen et al., 2002*; *Marshall et al., 2003*; *Van Neer, Jensen & Siebert, 2015*; *Hocking et al., 2017a*). Stunning prey included directly striking or tossing as well as indirect so-called "kerplunking" (slapping the water surface with tail to cause a startle response

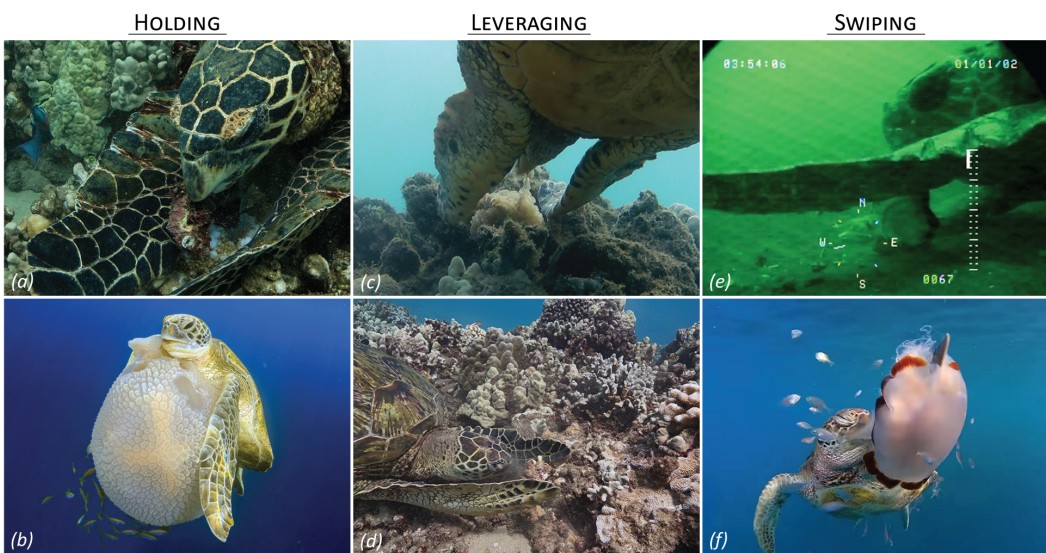

**Figure 1** **Limb use in marine turtle foraging.** (A) A hawksbill sea turtle holding a lobe coral (*Porites lobata*) to eat the black-brown protein sponge (*Chondrosia chucalla*) clinging to its surface in Kahekili, Maui USA, taken March 2010. (B) A green turtle holding a mosaic jellyfish (*Thysanostoma thysanura*) in the water column near the ocean surface in the Similan Islands, Thailand, taken June 2017 (© Rich Carey/Shutterstock.com). (C) A hawksbill sea turtle leveraging against the reef substrate to pry away a magnificent sea anemone (*Heteractis magnifica*). This was a frame grab from a video in Cook's Bay, Moorea, French Polynesia from June 2013. (D) A green turtle leveraging against the reef substrate to pry away bites of red macroalgae (*Amansia glomerata*) in Kahekili, Maui, taken October 2016. (E) A loggerhead sea turtle swiping the shell of an Atlantic deep-sea scallop (*Placopecten magellanicus*) while it consumes the edible tissue. This is a frame grab from a video in the mid-Atlantic Bight USA taken on July 2009 and available courtesy of the Coonamessett Farm Foundation (*Patel et al., 2016*). (F) A green turtle swiping the stinging jellyfish (*Cyanea barkeri*) in the water column at Hook Island, Queensland, Australia, taken June 2017. Image credits by the authors, save (B) © Rich Carey/Shutterstock.com and (E) Coonamessett Farm Foundation.

in prey) and was seen only in Delphinids (*Domenici et al., 2000*; *Gonzalez & Lopez, 2000*). Lobtail feeding (slapping water surface with tail during bubble-net feeding to corral prey together) is currently exclusive to humpback whales (*Megaptera novaeangliae*, (*Weinrich, Schilling & Belt, 1992*)). Sea otters (*Enhydra lutris*) demonstrated the most diverse and complex forms of limb-use for foraging, including pounding prey against tools (*Fujii, Ralls & Tinker, 2014*). To our knowledge, limb-use has not been documented in any other marine tetrapods, but future studies may reveal currently undescribed behaviors. Although publicly-available media have increased over time (*Kousha, Thelwall & Abdoli, 2012*), uploading images and videos on public platforms is not a universal practice. As a result, there may be other examples of limb-use that our surveys missed. However, the number of behaviors observed suggests it is still an effective method of documenting natural behavior.

We compared the prey type, relative prey size, and habitat across the marine tetrapods listed above to determine if distinct ecological factors promoted the development of limb-use (Table S1). However, limb-use behaviors were observed in a wide variety of conditions. Benthic feeders consumed bivalves, grasses, macroalgae, sponges, anemones, and hard corals. Epipelagic feeders consumed fish, jellyfish, and small marine mammals.

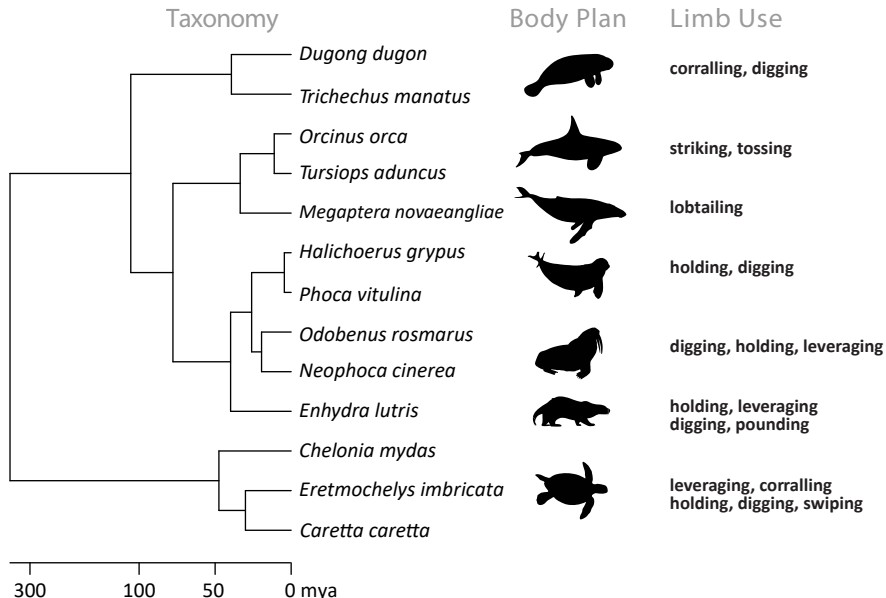

**Figure 2** **Evolutionary links between marine tetrapods known to use limbs while feeding and the diversity of body plans and types of limb use.** Silhouettes show a representative body plan for each family. Specific feeding behaviors are listed for each family.

Prey size often exceeded gape size (precluding whole consumption) but relatively smaller prey were also consumed, and included both mobile and sessile species. These factors may still be important factors at the species level, but did not remain constant across marine tetrapod families. Given the disparity of body forms and ecological niches observed across marine tetrapods, the range of ecological conditions in which limb-use is present is perhaps unsurprising. However, this diversity shows that a variety of conditions can lead to the expression of these behaviors, and provides additional support that many more species may use limb-use strategies that have not yet been documented in the literature.

The regular use of limbs for tasks beyond swimming may also promote the development of limb-use. As noted in *Hocking et al. (2017b)*, limb-use was more common in semi-aquatic mammals who may also use forelimbs for locomotion on land. In marine turtles, although predominately aquatic, females must return to land for nesting and use both fore- and hind-limbs to dig body pits and nesting sites (*Carr & Ogren, 1959*). Additionally, terrestrial basking by marine turtles is female-biased (*Van Houtan, Halley & Marks, 2015*). The wider range of flipper-use by female marine turtles may also result in a female sex-bias in limb-use for feeding. From our observations, all hawksbills we were able to age and sex were putatively determined to be females, but were unable to determine the sex of the other turtles due to limitations in the media. However, *Schofield et al. (2006)* observed digging by both sexes of loggerheads.

Limb mobility may play the largest role in the development of limb-use across marine tetrapods. Foreflipper mobility varies across marine tetrapods due to trade-offs for maneuverability, stability, and propulsion (*Fish, 2004*). *Taylor (1987)* suggested that

the constant need of foreflippers for locomotion and stability in the marine environment would limit their availability for other uses, including foraging. Although foreflippers used in propulsion have greater mobility compared to the foreflippers of taxa that use hindlimbs as the primary source of propulsion (*Fish, 2004*; *Kelley & Pyenson, 2015*), we found limb-use by species that used both forms of propulsion (Table S1). The limited mobility of foreflippers may prompt the use of tails in Delphinidae and Balaenopteridae cases, and may also explain the lack of limb-use by penguins and other cetacean families. Of the marine turtle foraging observations we report, all save one (Fig. 1B) involved foreflipper pronation movements. Figure 1B instead shows foreflipper supination while holding prey. Foreflipper pronations are the dominant mechanism marine turtles employ for swimming, crawling on land, excavating body pits for nesting, and aiding thermoregulation while basking (*Van Houtan, Halley & Marks, 2015*).

The origin of limb-use in marine turtles is currently unknown. Unlike other foraging strategies (such as lunge feeding), that can be analyzed via skull structure in extinct and extant species (*Motani et al., 2015*), it is currently unknown if there are any detectable physical predictors of limb-use that could be used for studying the origin of this behavior. Given the possible role of forelimb mobility on the presence of limb-use, muscular comparisons may be a future avenue of study as precedence for correlations between limb shape and ecological niche in turtles has been previously shown by *Joyce & Gauthier (2004)*. If limb-use is an (unselected) exaptation rather than an evolved trait, however, distinct morphological correlations may be more difficult to identify. Several species of terrestrial or semi-aquatic turtles have also been documented using their forelimbs to assist in processing food (*Davenport, Munks & Oxford, 1984*; *Lutz, Musick & Wyneken, 2002*), but the limbs of these species are not as specialized in shape as marine turtle foreflippers are for swimming (*Joyce & Gauthier, 2004*) and so this limb-use is similar to that seen in other terrestrial tetrapods (*Iwaniuk & Whishaw, 2000*). As marine turtles do not have opportunities for social learning, these behaviors either developed via independent trial and error, or are maintained as an innate behavior (*Lutz, Musick & Wyneken, 2002*). If these behaviors are innate, their presence in terrestrial turtles may support the hypothesis that this behavior was present in an ancestral turtle (*Joyce & Gauthier, 2004*). If this behavior was present when marine turtles evolved, approximately 120 million years ago, then limb-use may have been present in the marine environment almost 70 million years before all other extant marine tetrapods (*Bowen, Nelson & Avise, 1993*; *Cadena & Parham, 2015*; *Kelley & Pyenson, 2015*).

Although flipper morphology and foraging ecologies likely evolved via convergent evolution across marine tetrapods (*Kelley & Motani, 2015*; *Kelley & Pyenson, 2015*) it is unknown if limb-use evolved under the same processes. *Iwaniuk & Whishaw (2000)* showed that rudimentary limb-use likely first evolved in ancestral tetrapods but was subsequently maintained, developed, or lost in various lineages over time. It is, therefore, possible that the predisposition for this ancestral behavior was maintained as tetrapods returned to the marine environment and only manifests under appropriate modern conditions.

## CONCLUSIONS

The use of limbs to directly aid in foraging, while still relatively rare, is a strategy used by a variety of marine tetrapods. Despite being the oldest extant line of marine tetrapods, this is the first time such a wide range of limb-use has been described in marine turtles. We argue that these limb-use behaviors across marine tetrapods are limited by limb mobility and that the frequent use of forelimbs for other behaviors may promote the development of these feeding strategies. These observations provide additional insight into the diversity and possible evolution of limb-use behaviors.

## ACKNOWLEDGEMENTS

J Seminoff, N Pilcher, A Gentry, J Goldbogen, J Moxley, M Staedler, T Nicholson, M Murray, A Johnson, L Henkel and several anonymous reviewers improved earlier versions of this manuscript. The Coonamessett Farm Foundation provided underwater video.

### Funding

The authors received no funding for this work.

### Competing Interests

The authors declare there are no competing interests. Jessica Fujii and Kyle Van Houtan are employees of Monterey Bay Aquarium, Don McLeish is an employee of Hawaiian Hawksbill Conservation, and John Gaskell is an employee of Living Reef - Daydream Island.

### Author Contributions

- Jessica A. Fujii and Kyle S. Van Houtan conceived and designed the experiments, performed the experiments, analyzed the data, contributed reagents/materials/analysis tools, prepared figures and/or tables, authored or reviewed drafts of the paper, approved the final draft.
- Don McLeish, Andrew J. Brooks and John Gaskell performed the experiments, contributed reagents/materials/analysis tools, authored or reviewed drafts of the paper, approved the final draft.

### Data Availability

Fujii JA, Mcleish D, Brooks A, Gaskell J, Van Houtan KS. (2018). Limb-use by foraging marine turtles, an evolutionary perspective. Available at https://osf.io/b3dtq/.

### Supplemental Information

Supplemental information for this article can be found online at http://dx.doi.org/10.7717/peerj.4565#supplemental-information.

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
