# Peer review of "Limb-use by foraging marine turtles, an evolutionary perspective"

_PeerJ, doi:10.7717/peerj.4565_

## Round 0.1 · original submission · Major Revisions

Both reviewers are generally very positive about the potential contribution of this manuscript, however, both remain a bit concerned about the evolutionary interpretation of the behaviors that you are describing. You should clarify this point better. The reviewer 1 suggests "it would be easier to simply stick to what is known and only make some suggestions as to how old this might date back, and acknowledge the other pathways of how this process may have come to be".

The reviewer 2 pointed out that is not valid to state that the article represents the first observation of forelimb assisted foraging in adult marine turtles. It seems the forelimb use during foraging by mature sea turtles is a well-documented behavior and has been described in the scientific literature as early as the mid-1990’s. You have to address that as well.

I invite you to resubmit your manuscript after addressing all reviewer comments. When resubmitting your manuscript, please carefully consider all issues mentioned in the reviewers' comments, outline every change made point by point, and provide suitable rebuttals for any comments not addressed.

I look forward to reading an updated version of your paper.

·

Basic reporting

The text is clear and professionally put together. Literature is well reviewed and supports the manuscript arguments.

Experimental design

The research question is well defined and appropriate to the journal.

Validity of the findings

I believe the authors are a bit more speculative than needed at times, but this does not detract from a good read. The arguments are well presented and researched, and the conclusions are valid (although I suggest they acknowledge the other mechanisms through which these evolutionary processes may have occurred. See my detailed notes to the authors below.

Additional comments

This is a nice interesting read, and I think it would be interesting to others also. I think at times the authors read a bit too much into the findings, which could similarly have been written up by someone else as co-evolution processes – something along the lines of “imagine what a flipper might look like if there was no need to swipe or hold or dig or corral?” I am simply saying that while the authors argue that these are cases of exaptation, they could just as easily be of co-evolution. Given this, I think it would be easier to simply stick to what is known and only make some suggestions as to how old this might date back, and acknowledge the other pathways of how this process may have come to be.

I also think the authors made a big mistake in not reaching out to the sea turtle community for additional material for this thesis. Not everyone posts their video and imagery to the internet (I am one of those) and thus a very wide range or actions stands to have been lost. This does not mean the paper would not already have captured much of what has been seen, but it does point out a limitation in sampling.

Overall I think there are a few places where the MS could be tightened up, made more focused and less-speculative, but I also think this is a useful short addition to the body of knowledge on sea turtles in the public domain.

Some specific comments are made below:

Line 16: I am not so sure the word unexpected is correct here. There is no reason why an animal would not adapt if it could. So it is not so much that it is unexpected but rather uncommon or uncommonly documented.

Line 48 citation should be the first e.g. 2017a

Line 97 Word should be studies

Line 97 Might be best to clarify that this is in the mechanics of foraging rather than foraging in general – of which there have been many studies.

Sentence at lines 98-100 belongs in the Introduction and not in the Results section.

Line 102 If this has not been studied much, it might not be that it was not previously believed, It might just be something that previously received little attention – maybe because in today’s climate of endangered species there other issues are considered a higher priority to study and understand.

Line 103. There is no previously described foraging by green turtles described. It is not even mentioned above. The reader would need to know someone studied this, no?

Line 105 and elsewhere – the use of the word corralling is typically for something that moves – we corral bulls and horses. We do not corral non-motile sponges. Might this be better as collect, amass, round up, bunch up, or secure?

Similarly, the word capture is somewhat misleading as none of these are trying to escape. I suggest finding a better word here also.

Lines 107-108. How can the authors be certain these strategies are not necessary? On some reefs the organisms that the turtles seek are more cryptic than on other reefs, and turtles may be ‘forced’ to become adaptive. I am not at all certain this statement can be made with any validity.

Lines 109-117 could use a lot more explanation to make this more understandable to a public less-informed of these behaviours. Lobtail feeding… kerplunking… these could all use a short explanation. Understood that these show up In the table, but it would be worth some short inclusions of the less common ones in the text.

Lines 138-141 Here the authors correctly use the word ‘if’ because it is unknown if this is the case. But the Abstract reads as if this is conclusive. I suggest the Abstract be reworded to suggest only that this might be possible.

I am unsure of the meaning of lines 149-149. Given the huge disparity in animal forms, would this not be expected? What is being suggested there that is new and novel?

Lines 154-159 I believe the discussion really should be centered around exaptations in sea turtles and not delve into the why’s of dolphin and whale design. I do not see how this supports the discussion.

Lines 169-171 It is not possible to determine turtle sex simply by looking at the tail. These could easily have been immature, sub-adult males. I suggest the authors refrain from assumptions. It might be best to use the word putative here to indicate this is assumed but not known for certain.

Line 173 See my earlier comment on the use of the word unexpected.

·

Basic reporting

Overall, this paper is clearly written and I think the main point of the paper, that the use of forelimbs in foraging by marine turtles may have evolutionary relevance, is probably valid. Unfortunately, the evolutionary perspective needs to be much better demonstrated and the authors’ declaration that their article represents the first observation of forelimb assisted foraging in adult marine turtles is not valid. Forelimb use during foraging by mature sea turtles is a well-documented behavior and has been described in the scientific literature as early as the mid-1990’s.

Ex:

Limpus C., Couper P., & Read M. 1994. “The loggerhead turtle, Caretta caretta, in Queensland: population structure in a warm temperature feeding area”, Memoirs of the Queensland Museum, 37:195.

Houghton J., Woolmer A., & Hays G. 2000. “Sea turtle diving a foraging behavior around the Greek Island of Kefalonia”, Journal of the Marine Biological Association of the United Kingdom, 80:761-762.

Schofield G., Katselidis K., Dimopoulos P., Pantis J., Hays G. 2006 “Behaviour analysis of the loggerhead sea turtle Caretta caretta from direct in-water observation”, Endangered Species Research, 2:71–79.

In fact, forelimb assisted foraging in adult loggerhead turtles is so well known that it has been termed “infaunal mining” (See - Preen A.R. 1996. Infaunal mining: a novel foraging method of loggerhead turtles. Journal of Herpetology, 30, 94–96.) The authors should consider focusing the discussion of their paper on the unique foraging behaviors documented in their study (corralling, leveraging, holding, and swiping) or at least rephrasing/deleting the statements in the manuscript which imply the novelty of their observations.

Experimental design

Lines 67-68 – “We documented marine turtle foraging behavior from underwater surveys, web image and video searches (e.g., Google, YouTube, Vimeo, Flickr, Shutterstock)...”

Do the authors provide a list of these images and videos? Have the videos or images been uploaded to an online repository? Any visual data incorporated into the study should be accessible to other researchers and many of the URL’s for YouTube and Vimeo videos are not permanent and could expire/change at any time. The authors should consider contacting the parties responsible for producing any of the visual data used in the study and gaining permission to deposit the videos/images into a publicly accessible digital repository.

Line 92 – “Evolutionary relationships and divergence times are from Timetree.org”

Timetree.org should not be considered a primary source of information regarding the divergence estimates of sea turtles as the dates on Timetree.org are not based on the currently accepted best practices for such estimates. Recommended sources include:

Parham J., Donoghue P., Bell C. et al. (22 co-authors). 2012. “Best practices for using paleontological data for molecular divergence dating analyses.” Systematic Biology, 61:346–359.

Joyce W., Parham J., Lyson T., Warnock R. & Donoghue P. 2013. “A divergence dating analysis of turtles using fossil calibrations: an example of best practices.” Journal of Paleontology, 87, 612–634.

Validity of the findings

The authors do a good job of summarizing the different methods of foraging observed in extant chelonioid species, however, I think in order to say much about the presence of such behaviors in extinct species at the base of the chelonioid lineage, the authors needs to summarize quite a bit more data and I worry that this may be beyond the scope of the study but this is for the authors to decide. For example:

Lines 132-133 – “...it is currently unknown if there are any detectible physical predictors of limb-use that could be used for studying the origin of this behavior.”

Have the authors considered examining the anatomy of sea turtle forelimbs and those of other marine tetrapods known to engage in similar behaviors to see if there are any physical predictors associated with this type of limb usage? Vertebrate functional morphology and biomechanical inferences from comparative anatomy have been implemented across a multitude of extinct and extant species to assist in the determination of limb usage. Though foraging behavior would most likely be considered a biological role of a structure rather than its function, with the former being significantly more difficult to assess using traditional comparative anatomy, this may be an area the authors could investigate to further support their argument that no such physical predictor exists.

Line 137 – “...the limbs of these species are not as specialized as marine turtle foreflippers.”

This line of discussion is somewhat confusing as there is no discussion or citation provided regarding the morphology of aquatic, terrestrial, and marine turtle forelimbs as it pertains to the degree of specialization to individual ecologies. One potential source of such information may be:

Joyce W. and Gauthier, 2004. “Paleoecology of Triassic Stem Turtles Sheds New Light on Turtle Origins”, Proceedings of the Royal Society of London, Vol. 271, No. 1534.

Additional information on turtle forelimb morphology framed within an historical and ecological context can be found in:

Joyce W. 2007. “Phylogenetic Relationships of Mesozoic Turtles”, Bulletin of the Peabody Museum of Natural History, Vol. 48, p. 51.

Lines 137-138 – “This does, however, support the suggestion that this behavior was present in an ancestral turtle.”

This seems quite speculative as any discussion of paleoecology would need to rely heavily upon comparisons between the anatomy of fossil and extant taxa. While I agree that the presence of forelimb assisted foraging strategies in terrestrial, semi-aquatic, and marine turtles may serve as an indicator that these behaviors may have existed in some form in an ancestral turtle, this point should be better supported by data. Whether these data are purely anatomical in nature or a combination of ecological inferences based on an analysis of comparative morphology is up to the authors to determine. If the authors decide that a detailed anatomical discussion of turtle forelimbs is beyond the scope of the study, then perhaps these statements could be rephrased to better reflect what is currently known regarding the correlation between forelimb anatomy, biomechanics, and feeding ecologies in turtles (See Joyce and Gauthier, 2004).

Lines 138-139 – “If this behavior was present when marine turtles evolved, approximately 120 million years ago...”

This statement should be drawn from empirical data (see previous comments) rather than simple inference. The authors should also consider citing additional source regarding the earliest marine turtles (Ex. - Cadena E. and Parham J., 2015. Oldest known marine turtle? A new protostegid from the Lower Cretaceous of Colombia. PaleoBios, 32, 1-42)

Additional comments

Abstract

Lines 17-18 – “Here, we add marine turtles to the diversity of marine tetrapods known to use limbs for foraging...” Limb assisted foraging in marine turtles is well-documented. (See previous comments)

Introduction

Line 27 – “(Chelonioidea Oppel, 1811)” should be inserted following “Marine turtles”.

Lines 41-43 – Could the sentence regarding the presence of rudimentary limb-use for foraging be combined with the following sentence to better illustrate the connection between the two statements? Ex. “Rudimentary limb-use for foraging is observed in a range of terrestrial and aquatic taxa therefore, this behavior likely evolved in ancestral tetrapods...” This is only a suggestion assuming that I am correctly inferring that the authors are implying an evolutionary relationship between the terrestrial/aquatic and marine lineages.

Line 52 – “(Cheloniidae)” should be deleted. Cheloniidae refers only to hard-shelled sea turtles and excludes Dermochelyids (Dermochelys coriacea).

Lines 55-56 – “To date, however, marine turtle foraging mechanisms have received little attention.” This is not accurate. (See previous comments)

---

## Round 0.2 · accepted · Accept

Please follow up with the production team to make sure all text, figures, and tables are correct and ready for publication. Congratulations again.